# Vision and language representations in multimodal AI models and human social brain regions during natural movie viewing

**Hannah Small**
Department of Cognitive Science
Johns Hopkins University
Baltimore, MD 21218
hsmall2@jhu.edu

**Haemy Lee Masson**
Department of Psychology
Durham University
Durham, DH13LE, UK
haemy.lee-masson@durham.ac.uk

**Stewart H. Mostofsky**
Center for Neurodevelopmental and Imaging Research
Kennedy Krieger Institute
Baltimore, MD 21211
Department of Neurology, Department of Psychiatry and Behavioral Sciences
Johns Hopkins School of Medicine
Baltimore, MD 21205
mostofsky@kennedykrieger.org

**Leyla Isik**
Department of Cognitive Science
Johns Hopkins University
Baltimore, MD 21218
lisik@jhu.edu

**Editors:** Marco Fumero, Clementine Domine, Zorah Lähner, Donato Crisostomi, Luca Moschella, Kimberly Stachenfeld

## Abstract

Recent work in NeuroAI suggests that representations in modern AI vision and language models are highly aligned with each other and human visual cortex. In addition, training AI vision models on language-aligned tasks (e.g., CLIP-style models) improves their match to visual cortex, particularly in regions involved in social perception, suggesting these brain regions may be similarly "language aligned". This prior work has primarily investigated only static stimuli without language, but in our daily lives, we experience the dynamic visual world and communicate about it using language simultaneously. To understand the processing of vision and language during natural viewing, we fit an encoding model to predict voxel-wise responses to an audiovisual movie using visual representations from both purely visual and language-aligned vision transformer models and paired language transformers. We first find that in naturalistic settings, there is remarkably low correlation between representations in vision and language models and both predict social perceptual and language regions well. Next, we find that language-alignment does not improve a vision model embedding's match to neural responses in social perceptual regions, despite these regions being well predicted by both vision and language embeddings. Preliminary analyses, however, suggest that

vision-alignment does improve a language model's ability to match neural responses in language regions during audiovisual processing. Our work demonstrates the importance of testing multimodal AI models in naturalistic settings and reveals differences between language alignment in modern AI models and the human brain.

# 1 Introduction

As more and more powerful AI models are developed, a trend has emerged where models converge on the same latent dimensions. Recent work in NeuroAI shows that these same latent dimensions in vision models are the most brain-aligned, suggesting that they are universally extracted dimensions across artificial and biological intelligence [1]. This extends across modalities as well. Recent work found that multimodal "language aligned" training improves the neural match of visual model embeddings, and suggested that this multimodal advantage was specifically important for modeling social visual brain regions [2]. Some have found that even pure language model embeddings of image captions can predict visually-evoked activity in high-level visual areas [3, 4].

Other recent AI work has found that vision model representations of images and language model representations of their captions (without considering match to brain data) are also highly similar. This has led to an exciting recent proposal of a "platonic representation hypothesis" [5], where all models, regardless of modality, eventually converge on similar representations of an underlying world model.

Here, we ask what the implications of such a shared space would be for naturalistic multimodal processing. The evidence for shared vision-language representations mostly come from static scenes and language embeddings of their captions, but in real-world settings, simultaneous visual and verbal semantic signals do not always share a commonly referenced semantic space (e.g., the subject of speech is not always visible). Additionally, the processing of these two inputs are usually studied separately, either focusing on visual social signals (e.g., faces, biological motion, social interactions [6]), or social and semantic aspects of language [7].

In this work, we investigate naturalistic social processing and communication, which involves integrating converging but often disparate visual and linguistic input. We use a multimodal, CLIP-style transformer model as a tool to study this integration in the human brain. We first ask whether language, vision, and vision-language model representations are aligned with each other over the course of the movie. Then, we fit encoding models with the embeddings of pure vision, vision-language, and pure language models to predict human fMRI responses to an audiovisual movie (the first episode of the BBC television series Sherlock)(Figure 1). We combine this analysis with controlled experiments in the same subjects to specifically examine the responses of voxels that are sensitive to visual social interaction perception and auditory language understanding. We compare vision-language alignment in CLIP-style models to the human brain to ask whether joint vision-language model embeddings provide a better neural match than unimodal vision and language model embeddings.

# 2 Related Work

As mentioned above, one recent study has suggested that "language-aligned" visual embeddings produce a better match to human visual cortex in regions specialized for social perception [2]. This work, like the majority of NeuroAI vision studies, focused on neural responses to static scenes with relatively little social content. As others have suggested, naturalistic stimuli offer a promising path forward for studying social perception[8], which we explore here.

**Separate vision and language input**     To date, most encoding model studies of naturalistic fMRI data separately analyze either language responses from listening to stories or podcasts [9, 10, 11], or visual responses from watching silent movies [12, 13]. Recent work has identified shared semantic signals between vision and language (measured separately) along the border of visual cortex[14], as well as cross-modal prediction between vision and language via multimodal model embeddings [15], suggesting a shared conceptual space between vision and language.

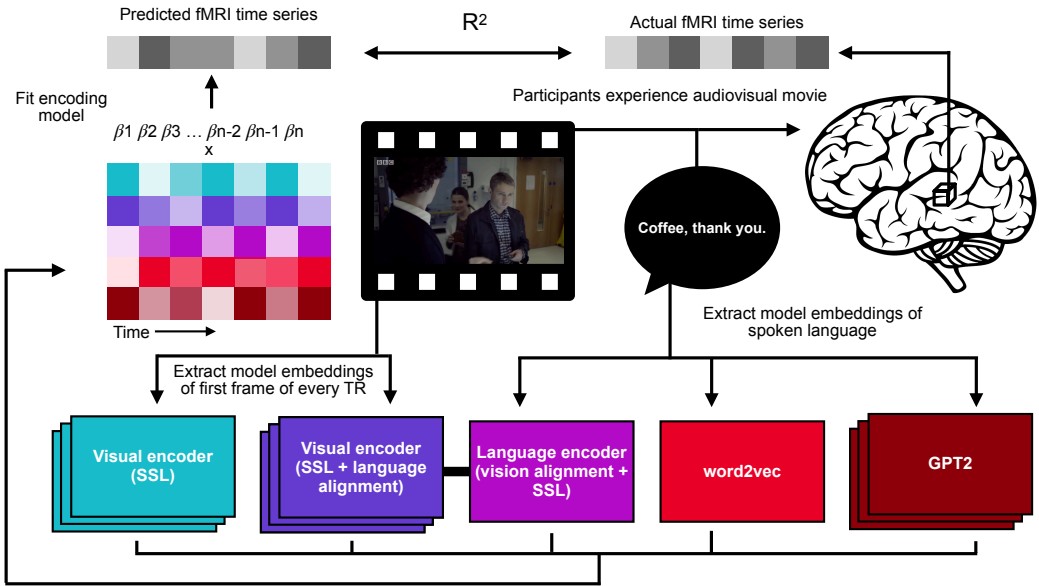

Figure 1: Encoding model approach. We recorded fMRI activity while participants (n=17) watched an episode of Sherlock. For each participant, we fit an encoding model with extracted vision model and language model embeddings from the movie frames and dialogue, respectively, to predict voxel-wise activity within an ISC mask.

**Multimodal NeuroAI with audiovisual input** Two recent studies have used multimodal models to predict neural responses to natural, audiovisual movies in fMRI [16] and sEEG[17], with somewhat conflicting results. In the first study, fMRI responses to the TV show Friends were predicted using embeddings from a multimodal (vision, language, and sound) model trained on 20 million YouTube videos [18]. They found that multimodal embeddings from the model did not predict brain responses better than the unimodal embeddings, but fine-tuning the model on a task that requires vision and language information resulted in more predictive multimodal embeddings in middle, but not later, model layers [16]. In the second study, neural responses, recorded using stereoencephalography (sEEG), to natural, audiovisual movies were predicted using embeddings from unimodal and multimodal models. This analysis revealed that multimodal models predicted better than unimodal models in about 12% of the sEEG sites[17].

In this work, we take a whole brain fMRI approach but focus in particular on the superior temporal sulcus (STS), a multimodal region critically involved in social perception that integrates multiple types of information [19]. By combining known experimental tasks targeting visual social interactions [20] and language [21] with a naturalistic audiovisual stimulus, we can compare multimodal AI models to relevant visual social and linguistic social brain regions.

## 3 Methods

### 3.1 fMRI experiments

Participants (n=17, neurotypical, ages 19-34, 10 female) watched the first 45 minutes of the first episode of the BBC series Sherlock. Participants were told to pay attention as if they were watching a television show they were interested in. We verified that they did pay attention with a recall task afterwards. They listened to the audio through MR-safe earbuds and watched the visuals on a screen reflected in a mirror inside the scanner. All participants also watched silent videos of point light walkers that were engaged in social actions and point light walkers performing independent actions to identify regions of the STS that are sensitive to social interactions, as in prior work [20]. A subset of participants (n=9) completed a language localizer, where they listened to audio of intact and degraded speech [21] to identify nearby language-selective voxels in the STS. All experiments were approved

by our local IRB. We used these tasks to identify motion, social interaction, and language selective regions in each participant.

Results included in this manuscript come from preprocessing performed using fMRIPrep 21.0.2 [22], which is based on Nipype 1.6.1 [23]. Full preprocessing details are available in Appendix A.1.1.

**Analysis of Localizers**    All localizers were analyzed with a General Linear Model (GLM). Data was mean-scaled prior to fitting the GLM and computing the first level contrasts. Each task model included regressors for all conditions along with confounds from fMRIPrep output, including the 6 rigid-body transformations, framewise displacement, and the aCompCor components from the combined white matter and CSF masks and the associated discrete cosine bases for high-pass filtering (cutoff of 128s).

From the point light walker task we identified social interaction selective voxels within posterior and anterior superior temporal sulcus (STS) using the contrast of interacting-independent dyads. We used a mask of STS and the temporal parietal junction (TPJ) [19], which we split into posterior and anterior portions, as in prior work [24]. We identified motion selective voxels within the FreeSurfer anatomical MT mask using the contrast of interacting & independent dyads. From the language task, we identified language selective voxels within previously identified temporal language regions [25] using the contrast of intact speech-degraded speech. For each contrast, we selected the top 5% most selective voxels across the corresponding parcels in left and right hemispheres.

### 3.2    Extracting unimodal and multimodal embeddings from AI models

**Models**    We used two self-supervised Vision Transformer (ViT) models, both trained on the YFCC15M (Yahoo Flickr Creative Commons) dataset, which is a subset of the YFCC100M dataset with English titles and descriptions [26]. One model was trained using a view-based self-supervised SimCLR-style contrastive vision objective [27] (referred to as SimCLR), while the other was jointly trained using the SimCLR-style objective and the CLIP objective, which aligns the image embeddings with language embeddings of their captions [28] (referred to as SLIP). We selected these models instead of the OpenAI CLIP model because their matched training data allows for more careful comparisons between purely visual and language-aligned models [29, 17]. We used the trained ViT-B versions of these models [30]. Both contain a 12-layer image encoder, with hidden dimension size of 768 and 12 attention heads per layer. The SLIP model additionally uses the smallest CLIP text encoder (maximum context length of 77) [30]. We do not have a language model that was trained with and without vision, so we compare the SLIP text encoder to two other "pure" language models, word level embeddings from word2vec [31] and sentence level embeddings from GPT-2 [32].

**Feature Extraction**    The fMRI acquisition time between fMRI images (repetition time or TR) was 1.5 seconds. To get features at this sampling rate, we passed the first frame of every TR (1.5s) through the visual encoders of SimCLR and SLIP and and extracted the activations from the attention output and patch embedding output of every layer, resulting in 24 feature spaces per model. We extracted the activations of each sentence of the spoken language content of the episode from each of the 12 transformer layers of the SLIP text encoder. As in prior work[14, 15, 9, 33, 34, 35], we resampled the embeddings to the fMRI sampling rate using a 3-lobe Lanzcos filter. This down-sampling procedure assumes that the neural response is the sum of responses [35]. As a point of comparison, we also extracted the embedding for every word from the word2vec model [31], which represents word-level co-occurence statistics. These vectors are similar to the text representations that language models receive as input. We resampled these embeddings using the same procedure as SLIPtext. Finally, we extracted the embedding of every sentence from every layer of the language transformer model GPT-2[32], with the exact same method used to extract embeddings from the SLIP text encoder. We specifically extracted the activations from the embeddings of the 12 transformer layers. We resampled each layer's embedding using the same resampling procedure as the rest of the language model embeddings.

**Dimensionality reduction with sparse random projection**    Each vision encoder feature space was 1921 time points x 197D x 768D, which we reduced to 1921 time points x 6480D (based on the Johnson Lindenstrauss lemma [36]) using sparse random projection (see details in Appendix A.1.2). For each layer of the SLIP text encoder (512D) and GPT2 (768D), and word2vec (300D), the

dimensionality was already less than 6480, so we did use sparse random projection on these feature spaces.

**Feature Space Similarity**    We measured the similarity between each feature space with Canonical Correlation Analysis [37, 38]. Given two feature spaces of any number of dimensions, this analysis finds latent dimensions of each feature space with maximal correlation with each other. For each pair of feature spaces, we projected each feature space to one latent dimension that was maximally correlated across both feature spaces. Specifically, we used L2 regularized CCA as implemented in the CCA-Zoo python library [39] (testing 5 log-spaced regularization parameters from $10^{-}5$ to 0) with nested cross validation (5 fold outer loop and 5 fold inner loop). The final similarity score was the correlation between the two feature space's test data projected onto their corresponding latent dimension.

## 3.3   Encoding Models

**Intersubject correlation mask**    All encoding model analyses were restricted to voxels with reliable stimulus-driven activity, measured using intersubject correlation (ISC) [40]. Intersubject correlation measures the shared neural responses across participants, which is taken to be the stimulus-driven signal. For each participant, we calculated the Pearson coefficient between their time series and the averaged time series of every other participant. This resulted in an ISC value for every voxel and every participant. We took the mean of Fisher z-transformed ISC values and then took the inverse Fisher z-transform of the averaged value. This is standard practice, as averaging the raw ISC values skews the mean downward[41]. All encoding modeling analyses were performed only in voxels with mean ISC > 0.15. The parcels used to find motion, social interaction, and language selective voxels are fully contained in this mask.

**Model Fitting**    We built a linear mapping between the feature spaces and the fMRI BOLD series for each voxel within the ISC mask with banded ridge regression models, implemented in the himalaya package [42]. Banded ridge regression allows each feature space to learn separate L2-regularization hyperparameters, which better accounts for differently sized feature spaces, prevents overfitting, and is especially useful for analyzing responses to naturalistic stimuli, where feature spaces are correlated [42]. We fit a hyperparameter for each layer of each model. We used 5-fold nested cross validation to fit the feature weights and select the regularization hyperparameters per feature space per voxel.

To account for temporal autocorrelation in the movie and fMRI data, we grouped the signal into windows of 20 TRs (30 seconds) before splitting into train/test sets in both the outer and inner cross-validation loops. To account for variable hemodynamic delays across cortex, all feature spaces were duplicated with time shifts of 1.5, 3, 4.5, 6, and 7.5 seconds. On every fold of the outer loop, the train set went through 5-fold inner loop regularization hyperparameter selection. The hyperparameters were sampled from a Dirichlet distribution and scaled by 25 log-spaced values between $10^{-}5$ and $10^{10}$. The best performing parameters, together with the estimated feature weights, were used to predict the fMRI response in the held-out test set of the outer loop. We measured prediction accuracy using the coefficient of determination $R^2$, averaging over the 5 folds to get the final value for each voxel in each participant.

**Variance decomposition**    We quantified the predictive contribution of each feature space compared to the rest of the feature spaces using the product measure[43, 44], which accounts for the correlation of the feature spaces. We use the product measure instead of the commonly used variance partitioning because it is more efficient for a large number of feature spaces [42]. The model prediction accuracy and product measure for each feature space was averaged across the 5 cross-validation folds. As in prior work [42], we calculated the proportion of total variance explained by each feature space by clipping negative values to 0 and then dividing the product measure by the joint model prediction to normalize the values to between 0 and 1.

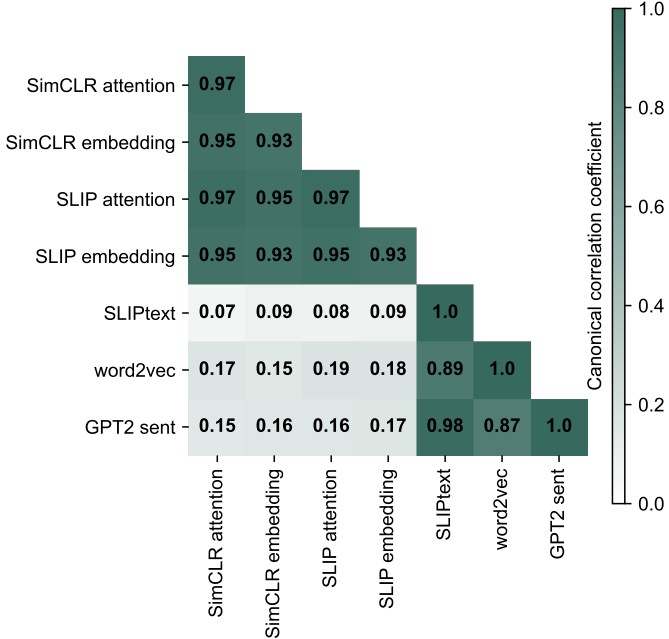

Figure 2: Feature space similarity of embeddings extracted from the movie. Each box represents the similarity between two feature spaces measured with CCA. The similarity of spaces from vision models, SLIPtext, and GPT-2 was measured layerwise. The average across 12 layers is shown. The full layerwise similarity matrix is available in Appendix Figure 8.

## 4 Results

### 4.1 Are vision model and language model representations aligned in a natural movie?

We first measured the similarity between vision model and language model embeddings of the frames and spoken language content in the movie . The vision embeddings of the first frame of each TR were extracted from the attention and embedding outputs of each layer of SimCLR and SLIP's vision encoder and the language embeddings of the spoken content from the movie transcript were extracted from SLIP's language encoder (SLIPtext), as well as word2vec and GPT-2. While there is strong similarity between different vision models (average $r$ range $0.93 - 0.97$) and language models (average $r$ range $0.87 - 0.98$), there is little similarity between the vision model and language model embeddings over the course of the movie (Figure 2). Interestingly, although SLIP's text encoder is trained to align the language embeddings with vision embeddings, it is less correlated with the vision model embeddings than word2vec or GPT-2 in this naturalistic context. The full layer-wise matrix, including the similarity between visual representations of different layers and models, is available in the Appendix (Figure 8). These results suggest that in this natural, social context, the vision and language DNN embeddings contain largely non-overlapping information.

### 4.2 Do CLIP-style multimodal AI models predict human STS responses during an audiovisual movie?

Next, we looked at the ability of different embeddings from the vision and language models to predict neural responses in visual and social brain regions. We focus our analysis in nearby regions of interest in the superior temporal sulcus (STS) that are involved in processing visual and linguistic social signals (posterior and anterior social interaction regions [20] and language regions[25]). We also compare our results to a visual control region, motion sensitive MT. We built encoding models with increasing language content, starting with the pure vision embeddings from SimCLR, next adding the language-aligned vision embeddings from SLIP, next the vision-aligned language embeddings from SLIPtext, and then language embeddings from word and sentence-level pure language models, word2vec and GPT-2, respectively.

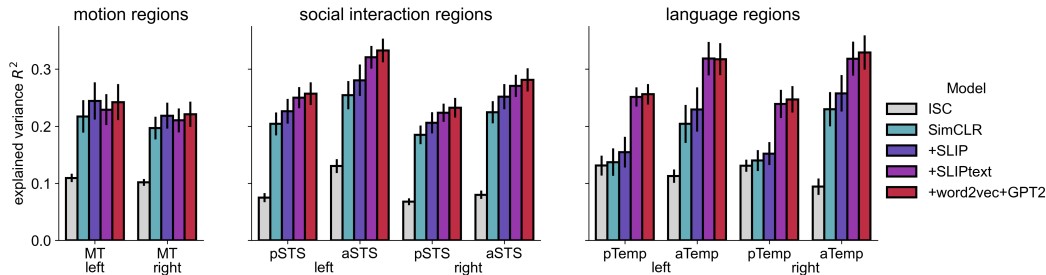

Figure 3: Encoding model performance of models with different groups of features, compared to intersubject correlation (ISC). Bars and error lines show the mean encoding model performance (coefficient of determination $R^2$)$\pm$ standard error of the mean (SEM) across participants.

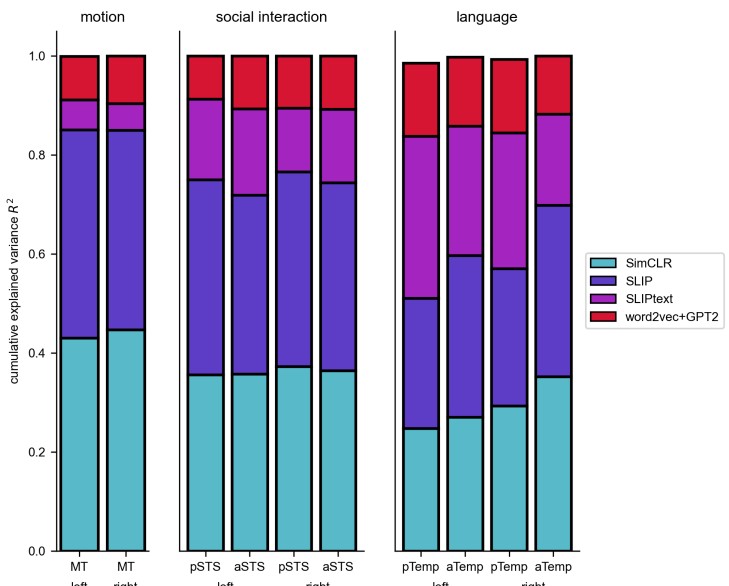

Figure 4: Predictive contribution of features in an encoding model with vision embeddings from SimCLR and SLIP and language embeddings from SLIP's language encoder, word2vec, and GPT-2. Each rectangle represents the variance explained by that feature space (all layers are added together when relevant), averaged across participant-defined regions of interest.

We find that all models perform significantly above chance and as well or better than the intersubject correlation in the same voxels (Figure 3). For both the social interaction and language regions, encoding model performance increases when we include features of the spoken language content from SLIP's vision-aligned language encoder. However, encoding model performance does not further increase when we include language features from word2vec and GPT-2 (Figure 3).

### 4.3 Does vision alignment allow language models to better predict human brain responses in the STS?

We also examined the predictive contribution of each feature space in each combined model. We find that the variance explained by SimCLR and SLIP's visual encoders remains the same with or without the presence of language features (Appendix Figure 9), suggesting that the vision model and language model embeddings contain non-overlapping information that correlates with neural signals in these regions. In the full encoding model, SLIP's language encoder explains relatively more variance than word2vec and GPT-2 in the left posterior social interaction region ($p < 0.05$) and left language regions ($p < 0.05$) (Figure 4 ), suggesting that these vision-aligned language features are better models of some social perceptual and language regions than unimodal word and sentence-level embeddings.

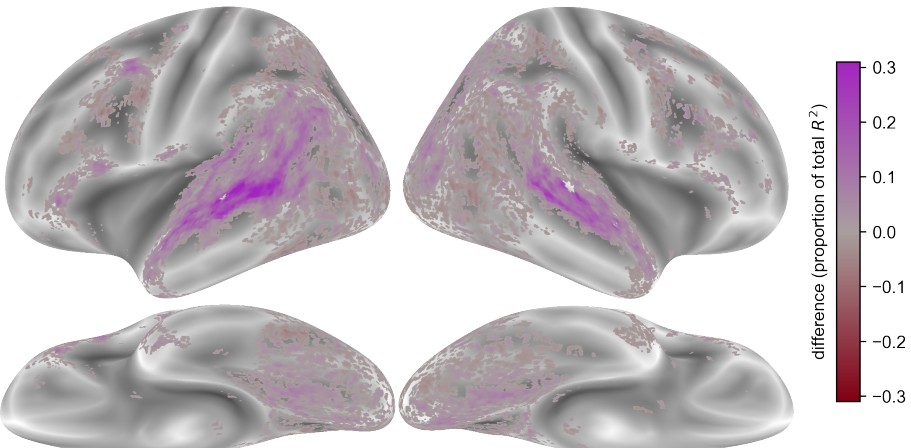

Figure 5: Group map of the difference in proportion of variance explained between SLIP's language encoder and GPT-2 in the full model, thresholded at difference of 0.01. Red indicates where GPT-2 explains more variance than SLIPtext and purple indicates where SLIPtext explains more variance than GPT-2. Group maps of SimCLR, SLIP, word2vec, and GPT-2 are available in Appendix Figure 10.

In a group whole brain analysis, we see that embeddings of the spoken language content from SLIP's language encoder predict brain activity throughout mid-to-anterior STS (Appendix Figure 10), and there is more variance explained by SLIPtext than GPT-2 in most of this region (Figure 5). Unsurprisingly, this effect is even stronger with a simpler language model word2vec (Appendix Figure 11). Together these results show that vision-aligned language embeddings predict social brain regions' multimodal responses to naturalistic stimuli.

### 4.4 Does language alignment allow vision models to better predict human brain responses?

We next asked whether language alignment improves brain predictivity in the STS, as it does for static stimuli in the ventral stream [2]. If language alignment creates visual representations that better predict the brain in social regions, we should see that a model trained with both self-supervised, view-based contrastive learning and language alignment (SLIP) explains a larger proportion of the variance in neural responses to the movie than a model trained on just self-supervised learning (SimCLR) in a social region.

We first compare the variance explained by SimCLR and SLIP in the full model when we simultaneously model the spoken language content of the movie (SimCLR+SLIP+SLIPtext+word2vec+GPT-2), and find no difference between these two models (Figure 4). This is also true when we do not control for the spoken language content (Appendix Figure 12) or fit each model individually (Appendix Figure 13 ), suggesting that language alignment does not improve neural predictivity in social perception or language regions.

In group whole brain analysis, we see that both SLIP and SimCLR predict brain activity throughout visual and higher level cortex (Appendix Figure 10). However, there is little difference between the proportion of variance that SLIP explains compared to SimCLR, with the exception of a SimCLR advantage throughout much of early visual cortex (Figure 6). Together, these results show that pure vision embeddings and language-aligned vision embeddings predict STS responses to naturalistic stimuli equally well.

Interestingly, we did see a much larger proportion of explained variance in both SimCLR and SLIP attention head outputs compared to the embedding layers (Figure 7), despite their similarly high correlation and matched ability to explain variance on their own (Appendix Figure 14). This advantage can be seen across cortex (Appendix Figure 15). Similar effects have been seen with language transformers [45]. The attention heads of transformers are known to be sensitive to positional information, which may explain why vision transformers do so well in predicting human cortex, which has been proposed to be scaffolded by visuospatial coding[46].

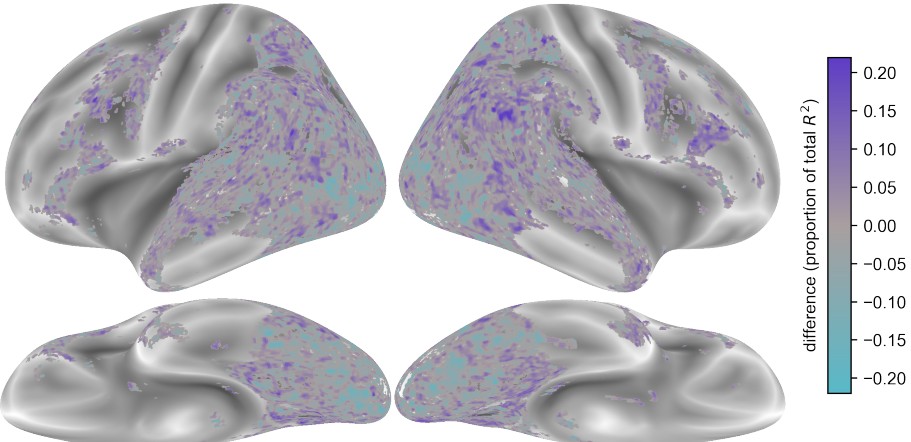

Figure 6: Group map of the difference in proportion of variance explained between SLIP and SimCLR's vision embeddings in the full model, thresholded at difference of 0.01. Dark blue indicates where SLIP explains more variance than SimCLR and light blue indicates where SimCLR explains more variance than SLIP. Group difference maps comparing the multimodal and unimodal language and vision embeddings. Group maps of SimCLR, SLIP, word2vec, and GPT-2 are available in Appendix Figure 10.

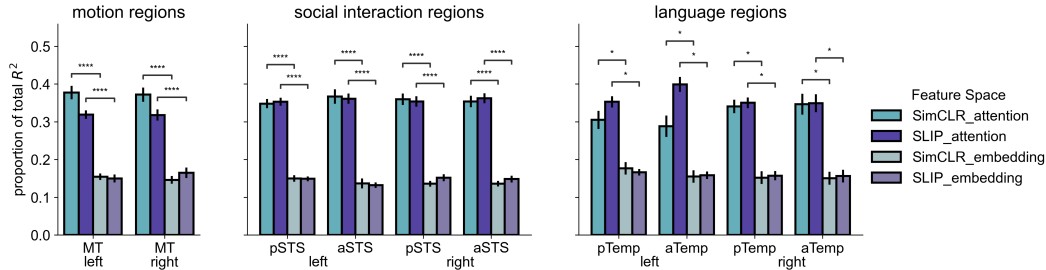

Figure 7: Predictive contributions of the attention head output and embeddings from SimCLR and SLIP (fit together in one model). The asterisks indicate a significant difference between the two feature spaces (Wilcoxon signed-rank paired difference test).

# 5 Discussion

In this work, we fit an encoding model with vision, multimodal, and language embeddings to predict voxelwise responses to a naturalistic audiovisual stimulus. We first showed that the vision and language representations learned by AI models, including multimodal, CLIP-style models, are not aligned over the course of a natural movie. We next showed that the SLIP model's vision-aligned language embeddings extracted from the spoken language content explain more variance than the language embeddings from unimodal sentence transformers like GPT-2 and word2vec, suggesting that vision-alignment may improve a language model's ability to match brain responses. Finally, we find that language-alignment does not improve the vision model's match to neural data. Overall these results suggest that one current state-of-the-art multimodal training approach, aligning images and their captions in a shared representational space, is not helpful for modeling neural responses to visual input, but may be helpful for modeling neural responses to linguistic input in naturalistic, audiovisual contexts.

The advantage of multimodal language models, however, should be interpreted with caution. Unlike the vision models, the multimodal and unimodal language models are not matched in terms of architecture, training data, or training objective, and there could be many reasons why SLIP's language encoder predicts better than GPT-2. A stronger claim could be made with a controlled comparison of larger and more powerful language models with the same architecture, dataset, and training objective, with and without vision. This is left to future work.

These results address a current gap in most NeuroAI research that has focused largely on single modality neuroimaging data and caption-based vision-language integration. Modeling naturalistic, social data, like movies, also raises questions about the extent to which shared "world models" are being learned by AI systems. One major limitation of this work is that it evaluates these image-caption trained models on more complex tasks of dynamic social perception. It might seem unsurprising that such models would fail to integrate vision and language in a human-like manner, however some past work has shown multimodal representational matches between these models and neural responses to naturalistic movies [17]. We also note that there are few good alternative models, as there is generally a dearth of models that produce human-like visual social perception [47]. Further, even with the current models, this work does call into question the idea that human-like social visual representations are an emergent property of static image-caption training.

The poor performance of these multimodal AI models in predicting visual brain responses to naturalistic stimuli calls for new approaches in modeling simultaneous vision and language. One approach may be to develop AI systems separately for relevant vision and language tasks and integrate them through specific interacting mechanisms, like cross-attention. Varying the points of vision-language integration could help determine what is sufficient to model multimodal brain responses. Recent work also suggests that dynamic information may be crucial for modeling social perception [47]. Thus, another promising direction is training on dynamic stimuli that have simultaneous vision and language signals in natural contexts. This may be enabled by recent high-quality, socially rich video datasets[48, 49] that could be used to train future multimodal models.

## 6 Acknowledgements and Disclosure of Funding

This material is based upon work supported by the National Science Foundation Graduate Research Fellowship under Grant No. DGE2139757 awarded to H.S. and a NIMH Exploratory/Developmental Research Grant Award under Grant No. R21MH129899 awarded to L.I. and S.M.

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

# A    Appendix / supplemental material

## A.1    Supplemental Methods

### A.1.1    fMRI preprocessing

The text of the following sections (Preprocessing of B0 inhomogeneity mappings, Anatomical data preprocessing, Functional data preprocessing) was automatically generated by fMRIPrep with the express intention that users should copy and paste this text into their manuscripts unchanged. It is released under the CC0 license.

**Preprocessing of B0 inhomogeneity mappings**    A total of 1 fieldmaps were found available within the input BIDS structure for this particular subject. A B0-nonuniformity map (or fieldmap) was estimated based on two (or more) echo-planar imaging (EPI) references with topup (Andersson, Skare, and Ashburner (2003); FSL 6.0.5.1:57b01774).

**Anatomical data preprocessing**    A total of 1 T1-weighted (T1w) images were found within the input BIDS dataset.The T1-weighted (T1w) image was corrected for intensity non-uniformity (INU) with N4BiasFieldCorrection [50], distributed with ANTs 2.3.3 [51], and used as T1w-reference throughout the workflow. The T1w-reference was then skull-stripped with a Nipype implementation of the antsBrainExtraction.sh workflow (from ANTs), using OASIS30ANTs as target template. Brain tissue segmentation of cerebrospinal fluid (CSF), white-matter (WM) and gray-matter (GM) was performed on the brain-extracted T1w using fast [52]. Brain surfaces were reconstructed using recon-all (FreeSurfer 6.0.1 [53]), and the brain mask estimated previously was refined with a custom variation of the method to reconcile ANTs-derived and FreeSurfer-derived segmentations of the cortical gray-matter of Mindboggle [54]. Volume-based spatial normalization to two standard spaces (MNI152NLin2009cAsym, MNI152NLin6Asym) was performed through nonlinear registration with antsRegistration (ANTs 2.3.3), using brain-extracted versions of both T1w reference and the T1w template. The following templates were selected for spatial normalization: ICBM 152 Nonlinear Asymmetrical template version 2009c [55] [TemplateFlow ID: MNI152NLin2009cAsym], FSL's MNI ICBM 152 non-linear 6th Generation Asymmetric Average Brain Stereotaxic Registration Model [56][TemplateFlow ID: MNI152NLin6Asym].

**Functional data preprocessing**    For each of the 7 BOLD runs found per subject (across all tasks and sessions), the following preprocessing was performed. First, a reference volume and its skull-stripped version were generated by aligning and averaging 1 single-band references (SBRefs). Head-motion parameters with respect to the BOLD reference (transformation matrices, and six corresponding rotation and translation parameters) are estimated before any spatiotemporal filtering using mcflirt (FSL 6.0.5.1:57b01774 [57]). The estimated fieldmap was then aligned with rigid-registration to the target EPI (echo-planar imaging) reference run. The field coefficients were mapped on to the reference EPI using the transform. BOLD runs were slice-time corrected to 0.7s (0.5 of slice acquisition range 0s-1.4s) using 3dTshift from AFNI [58]. The BOLD reference was then co-registered to the T1w reference using bbregister (FreeSurfer) which implements boundary-based registration [59]. Co-registration was configured with six degrees of freedom. First, a reference volume and its skull-stripped version were generated using a custom methodology of fMRIPrep. Several confounding time-series were calculated based on the preprocessed BOLD: framewise displacement (FD), DVARS and three region-wise global signals. FD was computed using two formulations following Power (absolute sum of relative motions [60]) and Jenkinson (relative root mean square displacement between affines [57]). FD and DVARS are calculated for each functional run, both using their implementations in Nipype (following the definitions by [60]). The three global

signals are extracted within the CSF, the WM, and the whole-brain masks. Additionally, a set of physiological regressors were extracted to allow for component-based noise correction (CompCor [61]). Principal components are estimated after high-pass filtering the preprocessed BOLD time-series (using a discrete cosine filter with 128s cut-off) for the two CompCor variants: temporal (tCompCor) and anatomical (aCompCor). tCompCor components are then calculated from the top 2% variable voxels within the brain mask. For aCompCor, three probabilistic masks (CSF, WM and combined CSF+WM) are generated in anatomical space. The implementation differs from that of Behzadi et al. [61] in that instead of eroding the masks by 2 pixels on BOLD space, the aCompCor masks are subtracted a mask of pixels that likely contain a volume fraction of GM. This mask is obtained by dilating a GM mask extracted from the FreeSurfer's aseg segmentation, and it ensures components are not extracted from voxels containing a minimal fraction of GM. Finally, these masks are resampled into BOLD space and binarized by thresholding at 0.99 (as in the original implementation). Components are also calculated separately within the WM and CSF masks. For each CompCor decomposition, the k components with the largest singular values are retained, such that the retained components' time series are sufficient to explain 50 percent of variance across the nuisance mask (CSF, WM, combined, or temporal). The remaining components are dropped from consideration. The head-motion estimates calculated in the correction step were also placed within the corresponding confounds file. The confound time series derived from head motion estimates and global signals were expanded with the inclusion of temporal derivatives and quadratic terms for each [62]. Frames that exceeded a threshold of 0.5 mm FD or 1.5 standardised DVARS were annotated as motion outliers. The BOLD time-series were resampled into standard space, generating a preprocessed BOLD run in MNI152NLin2009cAsym space. First, a reference volume and its skull-stripped version were generated using a custom methodology of fMRIPrep. The BOLD time-series were resampled onto the following surfaces (FreeSurfer reconstruction nomenclature): fsaverage. Grayordinates files [63] containing 170k samples were also generated using the highest-resolution fsaverage as intermediate standardized surface space. All resamplings can be performed with a single interpolation step by composing all the pertinent transformations (i.e. head-motion transform matrices, susceptibility distortion correction when available, and co-registrations to anatomical and output spaces). Gridded (volumetric) resamplings were performed using antsApplyTransforms (ANTs), configured with Lanczos interpolation to minimize the smoothing effects of other kernels [64]. Non-gridded (surface) resamplings were performed using mri_vol2surf (FreeSurfer).

Many internal operations of fMRIPrep use Nilearn 0.8.1[65], mostly within the functional processing workflow. For more details of the pipeline, see the section corresponding to workflows in fMRIPrep's documentation.

Data was smoothed with a 3mm FWHM kernel for subsequent localizer and encoding model analyses. Data was smoothed with a 6mm FWHM kernel for computing the intersubject correlation mask, which is in the recommended smoothing range [41].

### A.1.2   Sparse random projection

Sparse random projection projects a high dimensional feature space into a lower dimensionality feature space while preserving the pairwise Euclidean distance between points. The dimensionality of the lower dimensional space is determined using the Johnson-Lindenstrauss lemma and an epsilon specifying the amount of tolerated distortion [36]. Using the standard epsilon value of 0.1 and our sample size of 1921 time points, the Johnson-Lindenstrauss lemma outputs a target dimensionality of 6480 projections. These projections are randomly generated as a sparse matrix of nearly orthogonal dimensions. The feature spaces are projected onto this matrix using the dot product. The result is a 1921 x 6480 dimensional feature space, which is then used to predict neural activity in the encoding model. This pipeline has been used in several recent papers to speed up model fitting and to avoid overfitting [29, 47].

### A.2   Supplemental figures

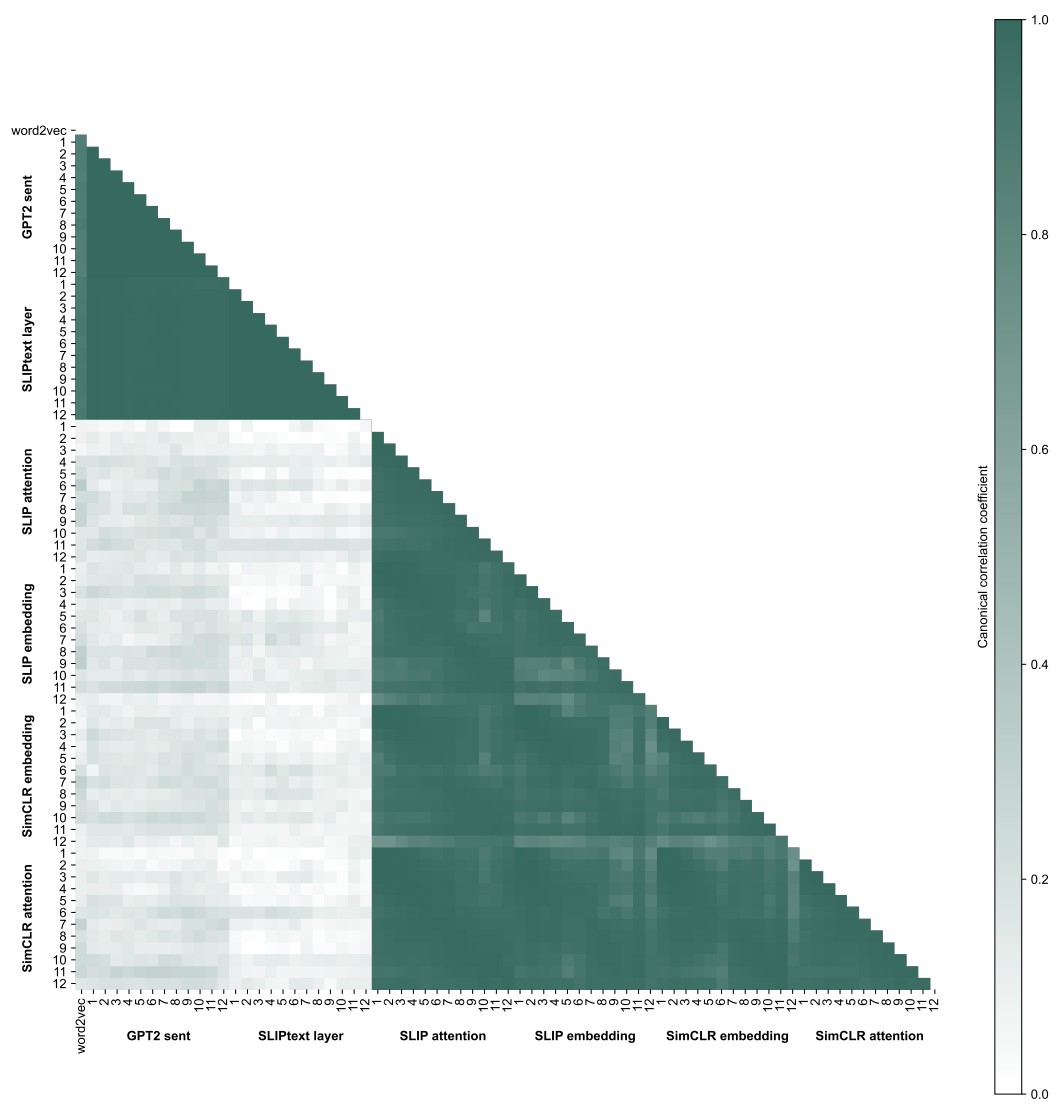

Figure 8: Similarity between vision and language model representations during a naturalistic movie.

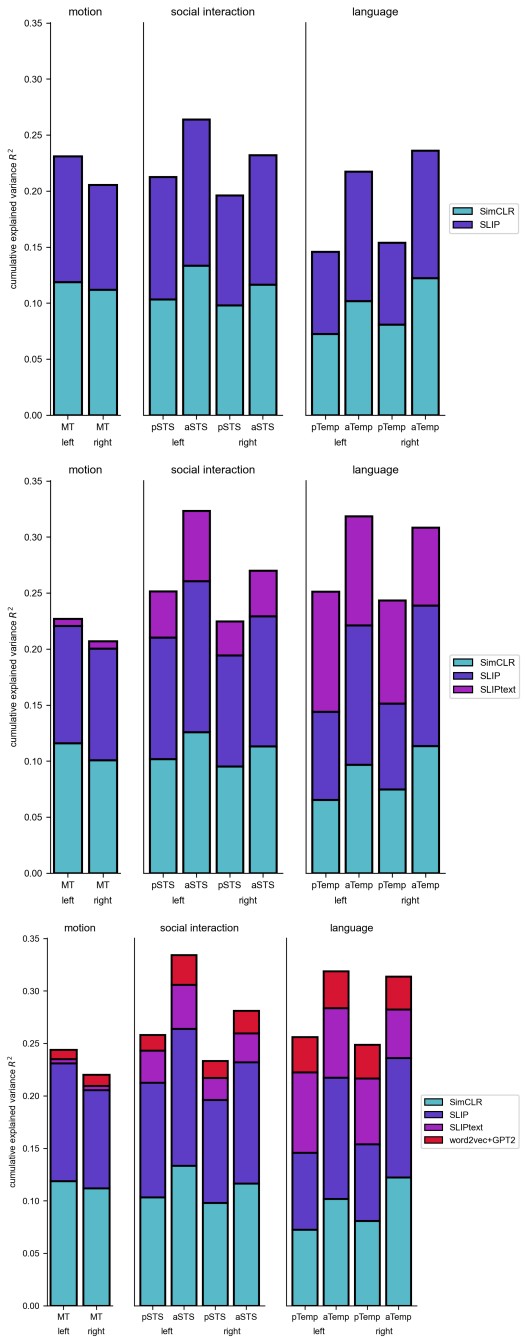

Figure 9: **Top**: Predictive contribution of features in an encoding model with vision embeddings from SimCLR and SLIP. **Middle**: Predictive contribution of features in an encoding model with vision embeddings from SimCLR and SLIP and language embeddings from SLIP's language encoder. **Bottom**: Predictive contribution of features in an encoding model with vision embeddings from SimCLR and SLIP and language embeddings from SLIP's language encoder, word2vec, and GPT-2 (reproduction of figure 4 from main text for visualization here). Each rectangle represents the variance explained by that feature space (all layers are added together when relevant), and averaged across participant-defined regions of interest.

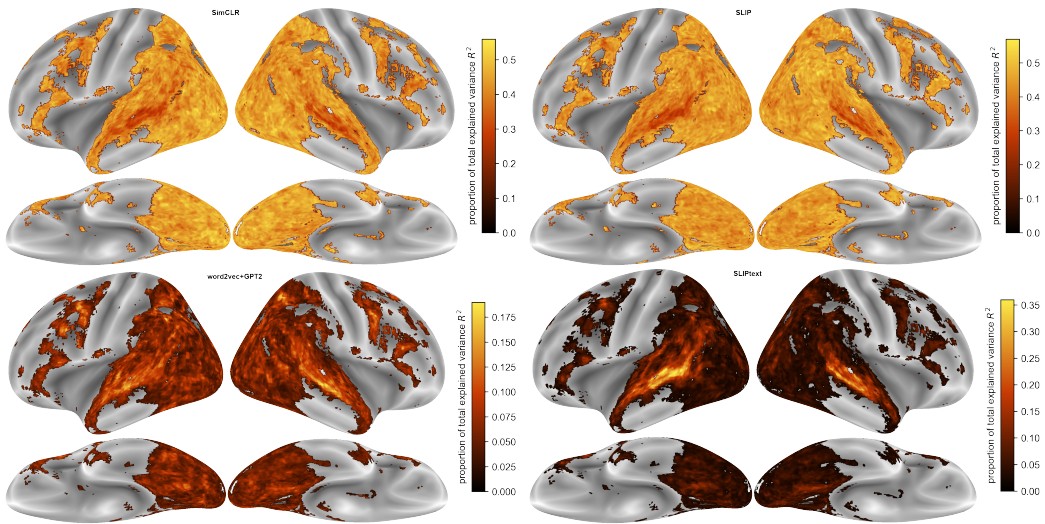

Figure 10: Group maps of the proportion of total variance explained by all layers of SimCLR, all layers of SLIP, all layers of SLIPtext, and all layers of GPT-2 + word2vec. All maps thresholded at 0.01.

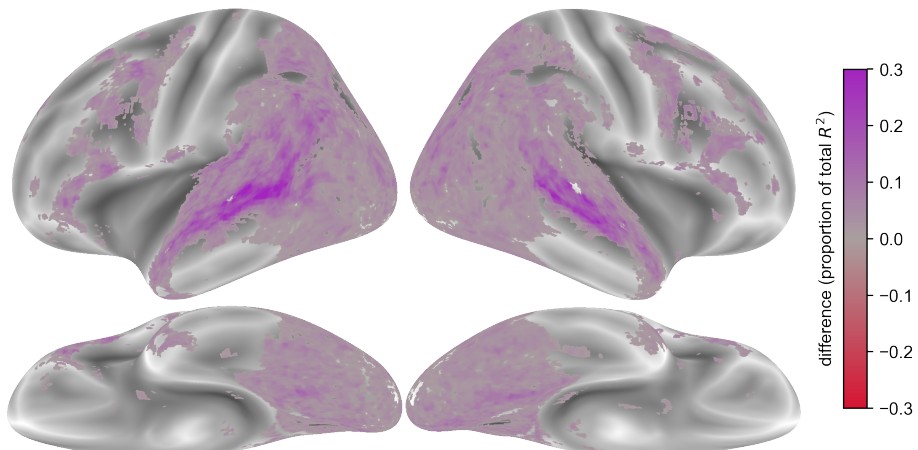

Figure 11: Group map of the difference in proportion of variance explained between SLIP's language encoder and word2vec in the full model, thresholded at difference of 0.01. Red indicates where word2vec explains more variance than SLIPtext and purple indicates where SLIPtext explains more variance than word2vec. Thresholded at 0.01.

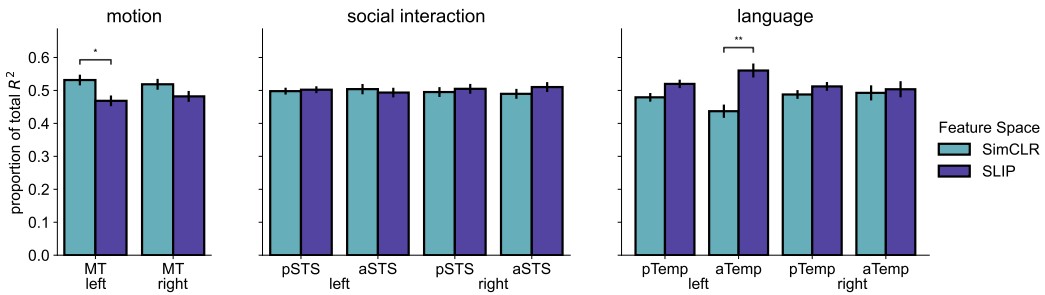

Figure 12: Proportion of variance explained by SimCLR and SLIP's vision embeddings when fit in one encoding model.

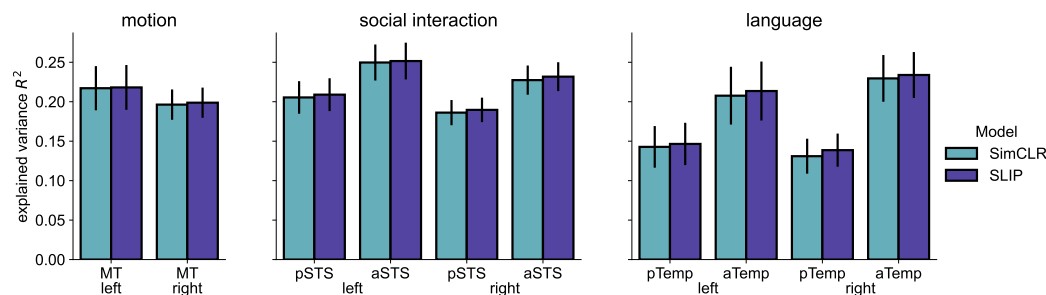

Figure 13: Encoding model performances of vision embeddings of just SimCLR and vision embeddings of just SLIP.

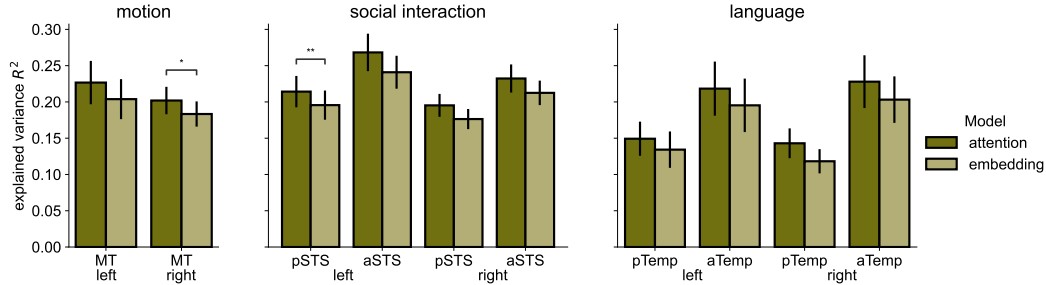

Figure 14: Performance of all attention head output and embeddings from SimCLR and SLIP.

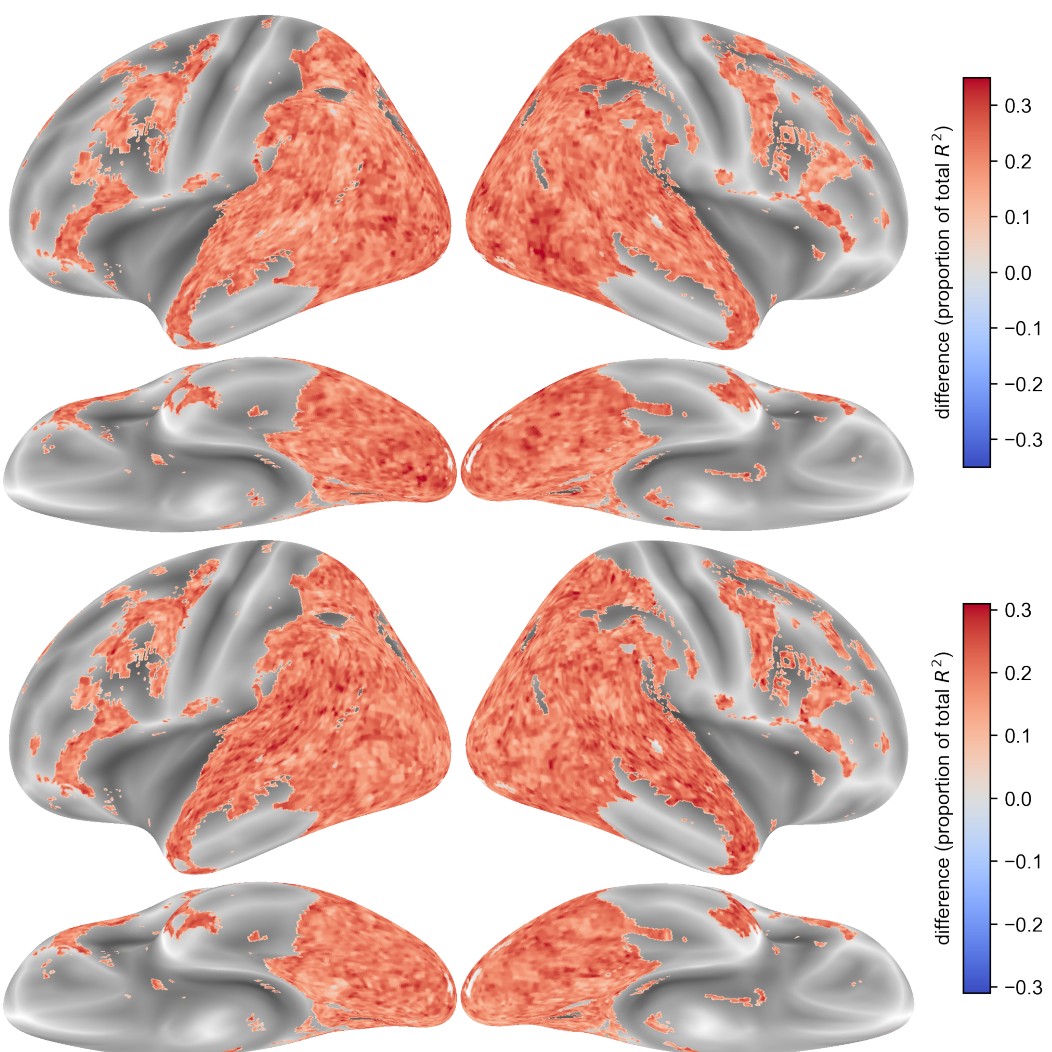

Figure 15: Group maps of the difference between all attention and all embedding layers in SimCLR (top) and SLIP (bottom). Pink indicates where attention predicts better than embeddings and green indicates where embedding predicts better than embeddings.

