# OpenReview forum: "Vision and language representations in multimodal AI models and human social brain regions during natural movie viewing"
_NeurIPS.cc/2024/Workshop/UniReps — UniReps_

### Official Review · Reviewer_LBQK · 2024-10-06
**Vision and language representations in multimodal AI models and human social brain regions during natural movie viewing**

**Rating:** 8
**Confidence:** 4

**Review:**

This article conducts a series of fMRI-related experiments using a deep learning approach, the results of which reveal that visual-linguistic alignment does not improve the model's matching of neural responses to visual, socio-perceptual, or linguistic regions, even though both visual and linguistic embeddings are excellent predictors of socio-perceptual and linguistic regions.

Possible drawback is that the article is not very easy to read as it contains more a priori knowledge but it does not prevent it from being an excellent article.

1.The article mentions multiple technical details (e.g., Fisher z-transform, ridge regression) when referring to data preprocessing and model fitting, but for readers who are not familiar with these methods.
2.Performance of the SLIP and SimCLR models in a visual-verbal alignment task, but no in-depth discussion of why the two models did not perform significantly differently in the prediction of social perceptual areas.

---

> ### Author Response · Authors · 2024-11-05
>
> Thank you for reviewing our paper! We have made several adjustments to address your concerns. We added some more background information on the methods we used and added some discussion of why the SLIP and SimCLR models might not perform differently in our results, as compared to previous work showing that language alignment improved prediction of high-level visual areas.

---

### Official Review · Reviewer_Qq5S · 2024-10-06
**Vision, not language, might contribute more towards predictivity of social region responses from current multi-modal DNN models**

**Rating:** 7
**Confidence:** 5

**Review:**

This study investigates how well current multimodal Deep Neural Networks (DNNs) predict fMRI activity in social regions responsible for visual and language processing. Previous evidence suggests that language-aligned vision models can predict visual social regions better than pure vision models, but they have been tested only with static stimuli without language components. The authors test unimodal (vision or language) and multimodal DNNs on their prediction of social regions (particularly superior temporal sulcus (STS)) during a naturalistic movie viewing task. The claims the paper forwards and related evidence are as follows:
> Claim 1: Vision model & language model embeddings are not aligned when processing natural movies.
- The vision embeddings are extracted only from the first frame of the TR, whereas the transcript from the whole TR (1.5s) is used to produce the language embedding. I believe the information available to the text & language models is largely unmatched, which might be a reason for the misaligned embeddings. I did not find enough evidence to support this claim.
- I disagree with the last line of this section stating that "in natural, social contexts, visual and linguistic signals contain largely non-overlapping information". Perhaps the implication here is that the signal in the vision/language DNN embeddings, please make this explicit.

> Claim 2: CLIP-style multimodal models predict human STS responses during natural movies.
- Encoding models mapping from different feature spaces (SimCLR, SLIP Vision, SLIP Text, Word2vec, GPT-2) to MT & STS predict explained variance better than ISC (Figure 3). The maximum contribution is from the vision embeddings (Figure 9), and language embeddings have a slight contribution (except in MT, as expected). Hence, I found enough evidence for this claim.
- It appears like all layers are pooled together to build the encoding model (Figure 4 caption), please state it explicitly.

> Claim 3: Vision-aligned language models predict the STS responses worse than language-only models during natural movies.
- From Figures 4, 5, 10, and 11, the prediction performance and contribution of SLIP-text (which is claimed to be vision-aligned) is lower than that of word2vec & GPT-2. However, the SLIP-text encoder was frozen during the vision training and only the projection to shared image-caption space was trained. Due to this, the degree of vision alignment of SLIP-text is unclear. Hence, I did not find enough evidence for this claim. Perhaps additionally testing other models with more vision alignment of their text encoders would provide more evidence to make the claim stronger.
- However, it is surprising to see that SLIP-text performs even worse than word2vec. Does this imply that vision-alignment (of the projection layer) decreases the STS predictivity? Comparing the predictive performance for SLIP text-only embeddings & (image-shared space) projection embedding can demonstrate more insights into this.

> Claim 4: Language-alignment does not improve STS prediction performance for vision encoders during natural movies.
- Figures 4, 6, 12, and 13 show that the SimCLR vision encoder & SLIP vision encoder perform equally well. While these claims are made for naturalistic (movie) stimuli, the vision models are fed only the first frame of the TR. Previous work demonstrated that language-alignment improves ventral stream predictivity (ref 2 from the paper), but the current study has diverging evidence from this. This could mean either the frame sampling strategy did not capture all the available visual information, or that there is no effect of language alignment on STS predictivity (as opposed to the effect on the ventral cortex). Hence, there is some evidence of this claim.
- Perhaps trying different frame-combining strategies, or citing studies showing no effect of such strategies, or comparison with video models would provide stronger evidence for this claim. If there is no effect of frame-combining strategies, then there seems to be enough evidence to support this claim.

Overall, this paper is novel in terms of investigating predictivity from multi-modal models to social regions in the human brain & collecting fMRI responses to naturalistic audiovisual stimuli. While the claims are not well supported, the paper nonetheless is useful for setting up methods for such complex comparisons, especially if the data and code are made public.

Suggestions & Questions:
1. Line 117, add the captions dataset & the relevant citation
2. Line 114, please cite YFFC dataset
3. Line 257, Typo: 7 -> Figure 7
4. Add the citations in A.1.1 functional data processing to the references & use the same citation style (numbers) as the main text
5. How was the threshold of the intersubject correlation (ISC) decided to be 0.15 (line 166)? Has this been used in previous work?
6. Intutively, adding more fMRI data (a variety of naturalistic stimuli) and considering more multi-modal models would make the claims stronger.

---

> ### Author Response · Authors · 2024-11-05
>
> Thank you for this comprehensive and helpful review. We have made the following edits to the camera ready version of the manuscript to address your concerns:
>
> - Claim 1: Vision-language model embedding alignment: Due to the inherent differences in timescale of vision and language inputs, there is not a perfect way to compare these two signals, so we turned to previous methods commonly used for vision (e.g., Lee-Masson & Isik, 2021) and language (e.g., Huth et al., 2016) encoding. We also note that the majority of our claims come from comparing multimodal and unimodal embeddings within vision or language domains (i.e., embeddings that are extracted in the exact same way, either vision to vision or language to language). Better aligning multimodal embeddings is an interesting area for future research.  We also updated the sentence you flag to more precisely reflect our results, thank you for bringing this to our attention.
>
> - Claim 2: Clarification on treatment of layers in the encoding model: Each layer of each DNN is specified as its own feature space in the joint model and we calculate the proportion of variance that each layer explains, then sum this over all layers of a model. We have added clarification in the methods.
>
> - Claim 3: Concern about vision-aligned language embeddings: Thank you for your suggestion to investigate the SLIPtext encoder. In looking into your suggestion, we realized that the  text encoder in the SLIP model is fully trained along with the projection to the shared caption-image space. This may allow the model to capture vision-aligned language information throughout the layers of the SLIP text encoder. Based on your suggestion, we re-ran our encoding models, using embeddings from every layer of SLIP’s text encoder, which is exactly what we do with GPT2. We find that SLIPtext actually does predict neural activity to an audiovisual stimulus better than GPT2 in left language region and left posterior social interaction regions. We have updated the figures with these new results and included some new discussion of these findings.
> The degree of vision-alignment of SLIPtext is a good question. In our feature similarity analysis, we find that SLIPtext is not as vision-aligned as word2vec or GPT2, yet it predicts neural activity better than either in left social interaction and language areas. Finding out why this is the case would be an interesting future direction of this work. More controlled comparisons between language networks trained with and without vision-alignment are needed to draw strong conclusions about this. We added a discussion of this in the text.
>
> - Claim 4: Concern about frame-combining strategies: Other work using naturalistic stimuli has shown there is no effect of frame-combining strategies on overall results (Garcia et al., 2024), suggesting that frame combination strategies do not explain the results we find. It is still possible that the temporal resolution of fMRI is not fast enough to distinguish between a language-aligned and non-language aligned visual representations in naturalistic contexts compared to static images, as in the previous work (Wang et al., 2023). Performing these analyses with neural signals with a higher temporal resolution would be an interesting area for future work.

---

### Official Review · Reviewer_eFN7 · 2024-10-06
**Solid methodological foundation, but limitations in model selection and training raise concerns about the validity of main conclusions**

**Rating:** 7
**Confidence:** 4

**Review:**

Sound methodology, but limitations in model selection and training constrain SLIPtext’s effectiveness (which is a key benchmark component of the study) and raise concerns about the validity of conclusions on vision-language integration for social perception.

For ROI selection, Social neuroscientist would argue why this paper approach the question regarding social perception in superior temporal sulcus (STS), not in temporal parietal junction (TPJ) and mPFC for more complicated social interaction. The stimulus for STS social perception localizer is quite simple (a visual social region), which is not usual stimuli in social study such as implicit social thoughts and theory of minds. However, I see good potential in this research direction of social perception and current paper provide a simpler approach that suits with the up-to-date technology.

The impact of Model Selection and training to results:

The whole brain analysis for SLIPtext shown in Figure 10 raise my concerns about the performance of the SLIPtext model. Its poor mapping to neural data suggests that the language component of SLIPtext is highly **constrained and suboptimal** for this task. The training method of SLIPtext, where the **text encoder is mostly frozen** and only the projection layer is learned, likely limits its ability to capture rich language information, explaining its weak mapping to broader brain regions. These factors combined lead to SLIPtext's poor performance compared to other pure language models like word2vec and GPT-2.

The conclusion about non-overlapping information between vision and language models might still hold, but it needs to be interpreted with caution. The non-overlapping nature could be more about the specific limitations of SLIPtext rather than a general property of vision-language models.

The another conclusion that SLIP model’s vision-aligned language embeddings are worse than pure word embeddings from word2vec, and SLIP model’s language-aligned vision embeddings are worse than pure vision embeddings from SimCLR may be **overstated**, due to the same reason.

More comprehensive multimodal models might produce different outcomes, while it's missing in the analysis nor discussed:

- While the choice of SimCLR and SLIP provides a clear comparison between visual-only and language-aligned models, it may limit the ecological validity of the study when comparing human learning processes. Multimodal models that integrate both modalities more thoroughly could better reflect human learning in social scenarios.

- To make a statement “Instead of using the current state-of-the-art multimodal training approach, aligning images and their captions in a shared representational space, is not helpful for modeling neural responses to naturalistic, audiovisual contexts.”, multimodal models only trained with separate visual/language encoders such as SLIP is not enough, other models with tightly coupled and integrated fine-tuning of Vision and Language inputs are necessary.

---

> ### Author Response · Authors · 2024-11-05
>
> Thank you for the positive and constructive assessment of your work. We have made substantial updates to the camera-ready version of the manuscript to address your concerns. Point-by-point responses are below:
>
> - ROI selection: Thank you for the suggestion to investigate theory of mind regions such as the TPJ and mPFC. These would be great areas for future investigation, but as theory of mind was not a major focus of this study, we did not include these localizers in our experiment.  While TPJ and mPFC are involved in mental state inference, multimodal social perception relies heavily on STS regions (McMahon & Isik 2023; Deen et al 2015). We clarified our emphasis on audiovisual social perception in the text.
>
> - SLIPtext model embeddings: Thank you for your suggestion to investigate the SLIPtext encoder. In looking into your suggestion, we realized that the text encoder in the SLIP model is fully trained along with the projection to the shared caption-image space. This may allow the model to capture vision-aligned language information throughout the layers of the SLIP text encoder. Based on your suggestion, we re-ran our encoding models, using embeddings from every layer of SLIP’s text encoder, which is exactly what we do with GPT2. We find that SLIPtext actually does predict neural activity to an audiovisual stimulus better than GPT2 in left language region and left posterior social interaction regions. We have updated the figures with these new results and included a new discussion of these findings.
>
> - Generalization of SLIP findings: Based on your suggestion and the results of the above analysis we have updated our conclusions to reflect the now higher performance of SLIPtext. However, we note that (unlike for visual embeddings) we still do not have a tightly controlled model comparison for multimodal versus unimodal language embeddings. We believe this is a fruitful area for future research. As you suggest, we also note that our findings are restricted to a specific model class. Throughout the text we have updated our claims to reflect our specific model comparison.

---

### Official Review · Reviewer_uvgP · 2024-10-06

**Rating:** 6
**Confidence:** 5

**Review:**

Summary: The paper designs multimodal brain encoding models for a movie viewing task, specifically focusing on social interactions. The paper focuses on the SLIP models and finds that (1) vision and language are often unaligned in these scenarios, (2) vision alignment hurts language model performance when predicting activity in the brain, and (3) language alignment has little effect on vision embedding alignment.

Strengths
1. The paper extends on lots of prior work focusing multimodal encoding models of the brain.
2. The paper uses specific models like SLIP which are controlled
3. The paper uses a new interesting dataset based on social interactions, which is unexplored
4. The paper also uses a GLM analysis to find ROIs

Weaknesses
1. The paper doesn't define what a TR is as far as I can tell so I'm having trouble understanding what data was fed to the encoding model and embeddings. This is important -- many of the results here will be influenced by what images and text were fed to all models. For example, in movies, vision is feature rich because visual stimuli are occurring all the time. But words are discrete. If you chose words and images that are far apart, they may not correspond. In [1], this was justified because it hurt alignment between the modalities. In this case, I think this could hinder results. I also couldn't find this in the appendix, so please include a discussion on what a TR is and how you selected image-text pairs to give to these models.
2. My familiarity with fMRI is low by 1.5s seems like a large window of time to run an encoding model over.

Questions
1. I wonder if using video models would help in this case. Somehow I wonder if the time dimension is what is missing here?

---

> ### Author Response · Authors · 2024-11-05
>
> Thank you for your constructive reviews. Please find responses below:
>
> - TR: Thank you for pointing out this omission. We have added a definition of TR/repetition time to the paper. To your points about continuous visual versus discrete words. We first note that the vision-language models are not trained using movie data, but instead pre-trained on image-caption pairs. For the movie encoding analyses, we agree it is tricky to time lock these two modalities that inherently have different rates of information change. As there is not a perfect way to compare these two signals, we employed methods commonly used in prior work for vision (e.g., Lee-Masson & Isik, 2021) and language (e.g. Huth et al., 2016) to model each signal.
>
> - For fMRI 1.5s is a relatively short TR. Unfortunately, fMRI is a slow, indirect measure of neural firing that only gives us a picture of brain activity every 1.5s. We take this into account using filtering methods described in the Feature Extraction section that are generally considered best practices (see point 1).
>
> - Indeed video models may improve match to performance, but since our goal was to use tightly controlled uni/multimodal vision models, which as far as we know are not available in video models. The low temporal resolution of fMRI (see point 2) somewhat mitigates the concern of temporal processing, though recent work has suggested video models may improve match to fMRI over image models (Garcia et al., 2024). Thus, we believe investigating multimodal video models is an important area for future work.

---

### Decision · Program_Chairs · 2024-10-10

**Decision:**

Accept

**Comment:**

In light of the reviewers' feedback and relevancy of the submission, we are pleased to accept this paper for presentation at UniReps 2024. We kindly ask the authors to incorporate the reviewers' suggestions and feedback in the final camera-ready version of the manuscript.